# Global Seroprevalence of Tick-Borne Encephalitis Antibodies in Humans, 1956–2022: A Literature Review and Meta-Analysis

**DOI:** 10.3390/vaccines12080854

**Published:** 2024-07-30

**Authors:** Patrick H. Kelly, Pingping Zhang, Gerhard Dobler, Kate Halsby, Frederick J. Angulo, Andreas Pilz, Harish Madhava, Jennifer C. Moïsi

**Affiliations:** 1Vaccines and Antivirals Medical Affairs, Pfizer US Commercial Division, New York, NY 10001-2192, USA; frederick.angulo@pfizer.com; 2Medical Affairs Evidence Generation Statistics, Pfizer Global Product Development Group, Collegeville, PA 19426-3982, USA; pingping.zhang@pfizer.com; 3Bundeswehr Institute for Microbiology, National TBEV Consultant Laboratory, 80937 Munchen, Germany; gerharddobler@bundeswehr.org; 4Vaccines and Antivirals Medical Affairs, Pfizer Biopharma Group, Tadworth KT20 7NS, UK; kate.halsby@pfizer.com (K.H.); harish.madhava@pfizer.com (H.M.); 5Vaccines and Antivirals Medical Affairs, Pfizer Biopharma Group, 1210 Vienna, Austria; andreas.pilz@pfizer.com; 6Vaccines and Antivirals Medical Affairs, Pfizer Biopharma Group, 75014 Paris, France; jennifer.moisi@pfizer.com

**Keywords:** TBE, tick-borne encephalitis, seroprevalence, *Ixodes*, *flavivirus*, *Orthoflavivirus*, *Flaviviridae*, tick

## Abstract

Despite the availability of tick-borne encephalitis (TBE) vaccines, the incidence of TBE is increasing. To understand the historical patterns of infection, we conducted a global meta-analysis of studies before December 2023 reporting human antibody prevalence against TBEV (TBE virus) among general or high-risk population groups stratified by country, collection year, serological method, and vaccination status. Pooled data were compared within groups over time by random-effects modeling. In total, 2403 articles were retrieved; 130 articles published since 1959 were included. Data were extracted from 96 general populations (117,620 participants) and 71 high-risk populations (53,986 participants) across 33 countries. Germany had the most population groups (21), and Poland had the most participants (44,688). Seven serological methods were used; conventional IgG/IgM ELISAs were the most common (44%). Four studies (1.7%) used NS1-ELISA serology. Between 1956–1991 and 1992–2022, anti-TBEV seroprevalence remained at ~2.75% across all population groups from “high-risk” areas (*p* = 0.458) but decreased within general populations (1.7% to 1%; *p* = 0.001) and high-risk populations (5.1% to 1.3%; *p* < 0.001), possibly due to differences in the study methodologies between periods. This global summary explores how serological methods can be used to assess TBE vaccination coverage and potential exposure to TBEV or measure TBE burden and highlights the need for standardized methodology when conducting TBE seroprevalence studies to compare across populations.

## 1. Introduction

Tick-borne encephalitis (TBE) is a major public health problem in several European and Asian countries. Despite the availability of two licensed TBE vaccines in Europe since 1992, following the launch of FSME-Immun^®^ in Austria in 1981 and Encepur^®^ in Germany in 1991, the incidence of TBE has increased in many European countries over the past two decades [1,2,3]. TBE is caused by infection with the TBE virus (TBEV), a single-stranded RNA virus in the genus *Orthoflavivirus*, most often spread through the bite of an infected *Ixodes* tick [4,5]. Less frequently, TBEV infections can also occur due to the consumption of TBEV-contaminated dairy products from TBEV-infected animals [6]. 

Globally, there are three main TBEV subtypes that broadly occupy distinct geographic regions: (1) TBEV-FE (Far East) is predominantly found throughout Asia, including Northern China and Eastern Russia; (2) TBEV-Sib (Siberian) is mostly distributed in Western Russia and Eastern European countries; and (3) TBEV-Eu (Europe) mainly circulates across the European continent [7]. Symptomatic TBEV infections in humans can be biphasic or monophasic and range in severity according to the TBEV subtype, with observed case fatality rates of <2%, 6–8%, and >20% in TBE cases caused by TBEV-Eu, TBEV-Sib, and TBEV-FE, respectively [5]. As a neurotropic virus, TBEV commonly causes clinical manifestations resulting from inflammation of the central nervous system, including meningitis and encephalitis. Although TBEV infection results in death in only 1–2% of hospitalized TBE patients, long-term sequalae among patients discharged from the hospital are common [5]. Based on a limited number of seroepidemiological surveys, previous TBEV infection among persons living in TBE-endemic areas is common but few infections are associated with severe illness and hospitalization, indicating that many TBEV infections are asymptomatic or clinically mild and unlikely to be diagnosed [8,9]. A study that followed a cohort of participants in Stockholm, Sweden who did not have TBEV at enrollment identified four persons with incident TBEV infections during follow-up, only one of whom developed TBE; symptoms in the other persons with new infections were not reported, so it is not known if they were asymptomatic or if they had mild, non-specific symptoms [9]. A cross-sectional study, which did not distinguish between asymptomatic and mild clinical TBEV infections, was conducted among residents of the Aland Islands of Finland, a highly endemic TBE area, and found that many had anti-TBEV antibodies, yet 80% of those with antibodies had no medical history of TBE [10]. The high proportion of mild illness and/or asymptomatic TBEV infections can lead to low clinical awareness and underreporting of TBE cases, particularly in areas with evolving risk. Differences in TBE surveillance systems between countries and public health agencies further complicate comparisons across populations, over time, or across geographic areas [11]. Countries with incomplete national testing programs or minimal capacities to test for anti-TBEV antibodies are also highly prone to surveillance gaps, limiting their ability to detect new TBE risk areas [11,12]. 

Seroepidemiological surveys can provide an estimate of the proportion of the population that has been affected or is currently infected by TBEV and thus provide information on (i) TBEV exposure rates in non-endemic populations such as travelers; (ii) TBEV seroprotection from vaccination based on neutralizing antibody testing; and (iii) the local TBE risk in a static population residing in a TBE-endemic area [13,14,15]. Several serological methods can be utilized in TBE serosurvey studies, with variable sensitivity and specificity, to estimate the disease burden or identify potential risk areas, including hemagglutinin inhibition (HI), indirect immunofluorescence (IIF), enzyme-linked immunosorbent assays (ELISA), and antibody neutralization tests (NT) [16,17]. Due to the high structural conservation and cross-reactivity of TBEV with other members of *Orthoflavivirus*, however, serological screening and diagnoses for past or current TBEV infections can be challenging among individuals living in or traveling from regions where other orthoflaviviruses, such as West Nile virus (WNV), dengue virus (DENV), and Zika virus (ZIKV), may be circulating [18]. Accurate regional estimates of the TBEV seroprevalence can also be difficult in areas with even modest vaccination coverage, since vaccine-induced anti-TBEV antibodies are indistinguishable from naturally acquired anti-TBEV antibodies (e.g., infected tick bite) in confirmatory assays like plaque reduction neutralization tests (PRNT). Recently, the establishment of an ELISA to detect antibodies against the non-structural protein 1 (NS1) specific to naturally acquired TBEV infections has enabled the assessment of TBEV exposure across both unvaccinated and vaccinated populations [19,20]. Seroepidemiological studies can now distinguish between anti-TBEV antibodies resulting from natural infection and those from vaccination using a conventional ELISA or NT coupled with the concurrent and reliable assessment of potential TBEV infections caused by infected tick bites based on the NS1-ELISA procedure within the same population [21]. 

In this review, we compile data on the prevalence of anti-TBEV IgM and IgG antibodies in humans from TBE seroprevalence studies from around the world throughout history and provide regional and country summaries of the data. We compare the anti-TBEV seroprevalence between general population groups and high-risk population groups over time before and after 1992, as the serological methods most used to detect anti-TBEV antibodies changed around this time. Lastly, we describe the different serological study methods used to measure anti-TBEV seroprevalence over time, to better understand the utility of seroprevalence data to assess TBE risks and burden of the disease in different population groups. 

## 2. Materials and Methods 

This study utilized systematic review methods. Analyses were conducted according to guidelines from pre-defined protocols, including clear inclusion criteria, search strategy, data collection process, synthesis methods, and data availability [22,23].

### 2.1. Systematic Literature Search

Two systematic literature searches were performed in PubMed and Embase to identify articles published before 30 June 2023 (“Search string one”) and 31 December 2023 (“Search string two”), with no restrictions on language or publication year. The exhaustive search string criteria for both search strings are described in Appendix A. Following the removal of duplicates, titles and abstracts were reviewed by three independent researchers to evaluate for relevance. Full-text reviews were then conducted by at least two independent researchers to confirm inclusion and exclusion. Articles not retrieved during the systematic literature search but considered for inclusion were reviewed separately and classified as identified via “additional sources” from personal communication, academic conferences, expert recommendations, citations during full-text reviews among articles included by the search string, or ad hoc database searches. 

### 2.2. Inclusion and Exclusion Criteria and Data Variable Extraction

Articles were included if they reported anti-TBEV seroprevalence data from non-hospitalized persons of any age. Articles that did not report the total number of individuals tested for anti-TBEV antibodies, failed to describe the serological detection method used in the study, or collected seroprevalence data potentially associated with other orthoflaviviruses due to cross-reactivity or alimentary TBEV infections were excluded. The following data elements from each study were extracted or calculated: publication year, continent, continental region, country, country region, study design (cross-sectional study with one population group, cross-sectional study with multiple population groups, or prospective study), population group (general population or high-risk population), description of participants (foresters, blood bank sera, etc.), vaccination status of participants (either as reported or inferred indirectly), age range of participants, selection method of participants (convenience or random), site of blood collection (e.g., blood banks, healthcare facility), sample collection years, serological method and manufacturer of serological test kits (if applicable), total number of participants in the study tested for anti-TBEV antibodies (denominator), total number of anti-TBEV seropositive participants (numerator), and proportion of anti-TBEV IgM and/or IgG seropositive individuals (anti-TBEV seroprevalence). Seroprevalence data were extracted as separate observations based on the region(s) where the study occurred (e.g., country, subnational region, study site) and in which samples were collected for each population group. 

### 2.3. Data Categorization and Analytical Structure

Studies of “general population” groups included those that analyzed samples from blood banks or healthy volunteer donors. If details were not provided about the samples or participants from the population group in the study, the data were considered “general population” groups. “High-risk population groups” consisted of participants with occupational exposure to ticks (e.g., forestry workers) or high self-reported exposure to tick populations. For regional data summaries, countries were grouped based on a qualitative assessment of similarly related TBE epidemiological factors and according to The World Factbook political geographies from the United States’ Central Intelligence Agency [24]. Epidemiological factors considered for regional country groupings included the endemicity of vector tick species (*Ixodes ricinus* and *Ixodes persulcatus*) and the circulation of TBE virus subtypes (TBEV-Eu, TBEV-Sib, and TBEV-FE). Countries were considered “endemic” for TBE if the country had one or more TBE-endemic areas (defined as the recurrent annual autochthonous transmission of TBEV infections in humans over multiple years) at the time (years) of sample collection. Data from TBE-“endemic” countries were further stratified as “high-risk” or “low-risk” for TBEV infection depending on whether the samples were collected from participants specifically in TBE-endemic or non-endemic areas, respectively, within TBE-endemic countries. If endemicity or risk stratifications could not be determined, expert opinion and other literature sources were sought. The data categorization of TBE-“endemic” countries, “high-risk” areas, and “low-risk” areas is shown in Appendix A.

### 2.4. Qualitative Assessment and Sensitivity Analysis

Studies included for data synthesis were qualitatively ranked as high-, moderate-, or low-quality based on the serological method used and the reported vaccination history of the study participants relative to the endemicity of TBE in the country and/or study region and when the samples were collected (Table 1). Specifically, qualitative assessments considered the endemicity of TBEV and other orthoflaviviruses in the study area, if/when a TBE vaccine was available to the public, and the potential vaccine coverage in the study population group. Studies met the “high” quality threshold if they (1) used NS1-ELISA serological method or (2) used an NT and confirmed medical and travel history in the study participants or enabled vaccination status verification based on the local vaccine introduction date (e.g., prior to 1976, when the first TBE vaccine was licensed in Europe) [2]. Studies were classified as “moderate” quality if they (1) used an NT but medical history could not be confirmed or (2) used a non-specific anti-TBEV serological method (ELISA, HI, or IIF) but medical and vaccination history could be confirmed and there was a low likelihood of the co-circulation of multiple orthoflaviviruses in the study region. Data were categorized as “low” quality if the studies used non-specific anti-TBEV serological methods and did not confirm the medical and vaccination history and/or there was a high likelihood of serological cross-reactivity with other orthoflaviviruses in the study area. For example, seroprevalence studies conducted in areas endemic for multiple orthoflaviviruses, such as Southeast Asia, were considered “low”-quality studies (or excluded entirely) if they did not perform confirmatory NT or NS1-ELISA serological testing and did not confirm vaccination or travel history in the study participants and reported elevated anti-TBEV seroprevalence with suspected cross-reactivity. The master datafile for the entirety of the data extracted for analyses is available for download (Appendix A).

### 2.5. Statistical Analyses

Statistical analyses were performed in SAS Studio (version 3.81), RStudio (version 1.4.1106), and GraphPad Prism 10.2.1. The numbers of population groups and study participants included for review from each country were mapped in QGIS (version 3.32.1) (Grüt, Böschacherstrasse 10A, Switzerland). The total number of population groups and total number of participants in each population group were calculated for each country, noting the beginning and end years of sample collection. The number of population group studies that utilized each serological method to generate seroprevalence data was recorded and calculated across the entire dataset with respect to each population group. To account for shifting trends and technical advancements in serological methods over the study observation period (1956–2022), temporal associations of anti-TBEV seroprevalence were measured in the population groups before and after 1992, which also corresponded to the public availability of two licensed TBE vaccines (FSME-Immun^®^ launched in Austria in 1981 and Encepur ^®^ launched in Germany in 1991). Data were analyzed based on the last year that samples were collected and tested for TBEV antibodies in the studies. If the sample collection years were not reported or unavailable for the study population groups, the publication year of the study article was used as a proxy for data pooling in the respective time (e.g., studies published before 1992 could be pooled in the “before 1992” period). Pooled anti-TBEV seroprevalence for each population group and for each geographic region was estimated via random-effects models and illustrated via forest plots with 95% confidence intervals, with the data quality inclusion/exclusion criteria specified (high, medium, and low). For temporal associations before and after 1992, the pooled anti-TBEV seroprevalence for each population group was compared across all countries, between TBE-endemic countries, and within “high-risk” TBE areas using data from high-quality and moderate-quality studies only. When comparing anti-TBEV seroprevalence among population groups between time periods, data were aggregated based on the year of sample collection. If a study reported anti-TBEV seroprevalence across multiple years, the observations were split proportionately across the reported years of collection. 

## 3. Results

### 3.1. Systematic Literature Review Characteristics

In total, 2009 articles were retrieved from the searches after duplicates were removed: 1895 were identified during the systematic literature searches and 114 were identified via additional sources (Figure 1A). A total of 130 articles met the inclusion criteria and were published in twelve different languages across eight decades (1959–2024) (Figure 1A,B; Appendix A). Sixty-one articles (46.9%) were from general population groups, 37 (28.5%) were from high-risk population groups, and 32 (24.6%) reported data from both (Figure 1B; Appendix A). Overall, thirty-five articles (27%) reported on more than one population group (Appendix A). Collectively, the 130 articles yielded TBE seroprevalence studies on 167 population groups (96 general population groups; 71 high-risk population groups) (Appendix A). Most of the studies (103/167; 61.7%) were cross-sectional with one population group, nearly one third (54/167; 32.3%) were cross-sectional with multiple population groups, eight (4.8%) were prospectively designed, and, in two, the study design was not described or could not be determined (1.2%) (Figure 1C; Appendix A). 

Anti-TBEV seroprevalence data were extracted from 171,606 participants (117,620 general population participants; 53,986 high-risk population participants) across 33 countries (28 countries with general population groups; 24 countries with high-risk population groups) and three continents (Figure 2; Appendix A). The country with the most population groups included in the review was Germany (21), followed by Italy and Poland (17 each), Sweden (13), and the Czech Republic (11), with ten countries represented by a single study (Figure 2). Poland and Germany had the largest numbers of participants, with 44,688 participants (26% of total) and 28,741 participants (16.7%), respectively, while eleven countries had fewer than 1000 total study participants across both population groups included in the review (Figure 2; Appendix A).

### 3.2. Summary of Study Cohorts, Study Design, Age, and Vaccination Status

The majority (129/167; 77.2%) of the population group studies were conducted in countries with at least one or more TBE-endemic areas and, most often (128/167; 76.6%), both children and adults participated in the studies (Appendix A). The age of the study participants was either not reported or unavailable in 39 (23.4%) of the studies. All but one study (166/167; 99%) measured the prevalence of anti-TBEV human IgG antibodies and 34 studies (20.4%) tested for anti-TBEV human IgM antibodies (Appendix A). In our qualitative assessment, we determined that 71 (42.5%) studies were high-quality, 53 (31.7%) studies were moderate-quality, and 43 (25.7%) studies were low-quality (Appendix A). 

Among all study participants, 82.7% (141,945) were unvaccinated, 2.3% (3968) were vaccinated or likely vaccinated, and 15% (25,693) had no reported vaccination history (Appendix A). The participants in the general population seroprevalence studies were most often healthy volunteers at healthcare facilities and blood banks (73%), but 3.6% of participants were patients seeking medical care unrelated to tick-borne diseases (Appendix A). The participants in high-risk population seroprevalence studies were predominately (85.6%) workers with occupational risks. Foresters were the most (43.7%) represented risk group, followed by farmers (21.1%), populations who reported “high exposure” to tick populations (25.4%), and military personnel (8.5%) (Appendix A). 

### 3.3. Serological Methods 

Seven different serological methods were utilized to identify IgM or IgG antibodies to TBEV: conventional ELISAs, a neutralization test (NT), a hemagglutinin inhibition assay (HI), indirect immunofluorescence (IIF), Western blot (WB), complement fixation (CF), and aNS1-ELISA (Table 2). The most common serological method was a conventional ELISA (43.6%), followed by NT (24.5%), HI (23.7%), and IIF (3.3%) (Figure 3; Table 2). An NS1-ELISA was used in four population group studies (1.7%; 2017–2023) (Figure 3; Appendix A). Across all serological methods, WB was the least used (<1%) (Figure 3; Table 2). Hemagglutinin inhibition assays were the most commonly (51.2%) utilized serological methods prior to 1992, and conventional ELISAs were the most frequent (58%) method in 1992 or after (Table 2; Appendix A). The majority of seroprevalence studies (101/167; 60.5%) used a single serological method to measure anti-TBEV seroprevalence, while the remaining studies used two (58/167; 34.7%) or three (8/167; 4.8%) different methods in eleven different combinations (Appendix A). Among the studies that used more than one serological method to detect TBEV antibodies, the most frequently used combinations were a conventional ELISA with NT (43.1%) and ELISA with HI (29.3%) (Appendix A). Detailed serological methods for the various combinations used in the studies and the number of serological methods used per decade are shown in Appendix A. 

### 3.4. Anti-TBEV Seroprevalence by Population Group and Time Period 

The prevalence of anti-TBEV antibodies was 2.9% (95% CI: 2.3–3.5%; I^2^ = 96%) in participants from general population groups and 4.6% (95% CI: 3.6–5.8%; I^2^ = 97%) in high-risk population groups (*p* = 0.002) (Figure 4A; Table 3). Sensitivity analyses including only high- or moderate-quality study data estimated a prevalence of 2.4% (95% CI: 1.9–3%; I^2^ = 96%) for general population groups and 3.6% (95% CI: 2.7–4.7%; I^2^ = 96%) for high-risk population groups (*p* = 0.051) (Figure 4A; Table 3). Before and after 1992, when conventional ELISAs became the most commonly used serological method in the studies, anti-TBEV seroprevalence decreased from 1.7% (95% CI: 1.2–2.2%; I^2^ = 86%) to 1% (95% CI: 0.6–1.4%; I^2^ = 77%) in the general population groups (*p* = 0.0011) and from 5.1% (95% CI: 4.1–6.3%; I^2^ = 93%) to 1.3% (95% CI: 0.8–2%; I^2^ = 83%) in the high-risk population groups (*p* < 0.001) (Figure 4B; Appendix A). In TBE-endemic areas, anti-TBEV antibody prevalence in high-risk population groups decreased from 5.1% (95% CI: 4.1–6.3%; I^2^ = 93%) to 2.2% (95% CI: 1.2–3.4%; I^2^ = 90%) before and after 1992 (*p* < 0.001); however, the prevalence of TBEV antibodies in general population groups did not substantially change between the two time periods (*p* = 0.484) (Figure 4C; Appendix A). Finally, across all population groups in high-risk areas from TBE-endemic countries, anti-TBEV prevalence was similar between the two time periods, from 2.8% (95% CI: 2.3–3.3%; I^2^ = 91%) to 2.7% (95% CI: 2–3.5%; I^2^ = 77%) (*p* = 0.4582) (Figure 4D; Appendix A). 

### 3.5. Western and Central Europe

There were 87 TBE seroprevalence population group studies (49 general population group studies and 38 high-risk population group studies) in Western and Central Europe (Appendix A). Samples from 101,185 participants (60,400 general population participants and 40,785 high-risk participants) were collected between 1959 and 2021 in thirteen countries. Seventy-two (82.7%) studies were conducted in TBE-endemic countries (21 in Germany) and fifteen were conducted in countries with no TBE-endemic areas at the time of sample collection (ten in Italy, three in the Netherlands, and one each in Portugal and Belgium). The aggregated anti-TBEV seroprevalence across the region was 2.7% (95% CI: 2.1–3.3%; I^2^ = 94%) for general populations and 4.6% (95% CI: 3.4–6%; I^2^ = 97%) for high-risk populations (Appendix A). Fourteen studies (nine in Italy, two in Germany, and one each in Liechtenstein, Poland, and Belgium) found no seropositive study participants [25,26,27,28].

Seventeen studies were conducted in Poland (1965–2021), which included 20,999 general population participants with reported anti-TBEV seroprevalence between 0 and 8.5% across eight studies and 24,389 high-risk population participants that reported a range seroprevalence between 6.6 and 87.5% within nine studies. The high-risk population group study that reported anti-TBEV seroprevalence of 87.5% was performed in foresters (n = 32) working in Bialowieza National Park between 2003 and 2007; this was the highest observed anti-TBEV seroprevalence across any study included in the review [29]. Six studies were conducted in Slovakia between 1959 and 2016, with anti-TBEV seroprevalences ranging between 2.8 and 42.4%. Four studies between 1959 and 1964 were performed in the TBE-endemic regions of Jarok, Kostolany, and Nemcinany, while another more recent study in the eastern TBE-endemic part of the country measured 2.8% TBEV antibody prevalence in a general population cohort between 2014 and 2016 [30,31,32,33]. Eleven studies were conducted in the Czech Republic (1963–2018), one of which used the NS1-ELISA serological method to conduct national seroprevalence studies in vaccinated military members and found seroprevalence of 6.6% [34]. Notably, a recent study (2001–2014) across a national general population cohort reported and confirmed via NT that 71/270 (26.3%) individuals had antibodies against TBEV [35]. Seventeen studies were from Italy, and most (10/17; 58.8%) were performed in non-endemic (“low-risk”) TBE areas, including Campania, Grosseto, Fondi, Friuli, Tuscany, Turin, and Venezia, where the seroprevalence for TBEV antibodies ranged between 0 and 5.7%.

Another study included 17,319 healthy blood donor participants from 17 provincial voivodeships between 1965 and 1967 and found 1.9% anti-TBEV seropositivity via HI. A follow-up cross-sectional study across the same regions in 21,425 foresters between 1971 and 1972 found a TBEV antibody prevalence of 6.6% [36,37]. Most studies in Germany (76.1%) occurred in southern states (Bavaria (Bayern), Baden-Württemberg, Rhineland-Palatinate, and Saarland) endemic for TBE. The study in Germany with the highest reported anti-TBEV seroprevalence described a rate of 33.4% in 686 farmers and foresters from Lower Franconia, Germany between 1965 and 1967 [38]. One study in Baden-Württemberg used NS1-ELISA serology and determined anti-TBEV IgG antibody prevalence of 5.7% in 2200 blood bank serum samples; this was nearly seven-fold higher than the 0.8% anti-TBEV seroprevalence reported in Baden-Württemberg by another general population group study 40 years prior [21,39]. 

Five countries in Western and Central Europe were represented by a single article: Austria, Belgium, Liechtenstein, Portugal, and Slovenia. The lone study in Austria was conducted in 1961 among 1412 subjects from the Neunkirchen district and found anti-TBEV seropositivity of 15.5% based on a HI assay and CF and confirmed via NT [8]. A national study in Portugal found that 0.2% (3/1649) of the general population study participants had antibodies against TBEV [40]. The mean seroprevalence of TBEV antibodies for each population group in Western and Central Europe is summarized via a forest plot in Appendix A. 

### 3.6. Scandinavia

Scandinavia was represented by 23 seroprevalence population group studies (fourteen in general population groups; eight in high-risk population groups) between 1956 and 2022 from three countries: Sweden (13), Norway (7), and Denmark (3) (Appendix A). All but one (95.6%) study were conducted in TBE-endemic countries/regions, and all but two (87%) of the studies were in areas considered “high-risk” for TBE. The aggregated anti-TBEV seroprevalence across the region was 1.9% (95% CI: 0.9–3.2%; I^2^ = 93%) for general populations and 3.4% (95% CI: 1.1–6.7%; I^2^ = 95%) for high-risk populations (Appendix A). Seventeen (73.9%) of the studies in Scandinavia used two or more serological methods to detect TBEV antibodies, which was the highest of any region. 

The three studies in Denmark were published over a sixty-year period, from 1962 to 2023. The most recent population group study utilized NS1-ELISA serology in a national serosurvey of TBEV antibodies across five geographic regions, estimating a seroprevalence at 0.1% [41]. A study in Danish foresters (n = 39) in 1960 from the island of Bornholm had the highest reported anti-TBEV seroprevalence at 28.2% [42]. Eight (61.5%) of the thirteen studies in Sweden were in high-risk population groups. The oldest TBE seroprevalence study included in the review was conducted in 1956 in the greater Stockholm region and identified one (1.3%) anti-TBEV seropositive subject [43]. Finally, the seven studies in Norway were all in general populations concentrated in the southern coastal regions of the country between 1973 and 2019 and had seroprevalence ranging between 0 and 19.6%. A forest plot summary of the mean seroprevalence of TBEV antibodies in each population group in Scandinavia is shown in Appendix A. 

### 3.7. Northwestern Eurasia

In Northwestern Eurasia, we identified fourteen seroprevalence studies with anti-TBEV serology data from three countries between 1958 and 2022 (Appendix A). The aggregated anti-TBEV seroprevalence across the region was 7.4% (95% CI: 2.7–14.1%; I^2^ = 100%) for general populations and 11.8% (95% CI: 5.7–19.6%; I^2^ = 93%) for high-risk populations (Appendix A). Studies in Northwestern Eurasia were least likely (21.4%) to use confirmatory serological methods specific to TBEV (NS1-ELISA or NT) and the majority (64.3%) only used one serological method overall. 

Two of the six studies in Finland were follow-up studies of recently tick-bitten participants, which collectively identified four (0.6%) anti-TBEV seroconverted individuals [44,45]. The five studies in Russia were conducted in Northeastern Siberia between 1986 and 2012 and the Western Urals between 1965 and 1989 [46,47,48,49,50]. Collectively, they had antibody prevalence between 0 and 50.7%. The mean seroprevalence of TBEV antibodies for each population group in Northwestern Eurasia is shown as a forest plot in Appendix A.

### 3.8. Southeastern Europe

We identified 26 seroprevalence studies in Southeastern Europe (14 general populations and 12 high-risk populations), conducted between 1962 and 2022 in seven countries (Appendix A). The aggregated anti-TBEV seroprevalence across the region was 3.3% (95% CI: 2.1–4.6%; I^2^ = 91%) for general populations and 4.9% (95% CI: 1.9–9.1%; I^2^ = 97%) for high-risk populations (Appendix A). 

Nine studies between 1972 and 2020 were performed in Romania (the most in the region), with anti-TBEV seroprevalence that ranged from <0.01% to 30.1%. Serbia had the second-largest number of studies in Southern Europe (8), with seroprevalence ranging from 0 to 15%. For Croatia, one high-risk population study in foresters in 2000 found 1.4% anti-TBEV seropositivity, and a separate study in foresters and healthy controls in 2010 measured 4.4% [51,52]. Two population group studies each were included from Greece and North Macedonia, and Bulgaria and Bosnia and Herzegovina were each represented by one study. The lone study in Bulgaria had an anti-TBEV seroprevalence of 0.6% in a nation-wide general population cohort [53], and the high-risk population group study in Bosnia and Herzegovina found a seroprevalence of 0.4% in a United States military cohort stationed in the country in 1996. A forest plot summary of the mean seroprevalence of TBEV antibodies in each population group in Southeastern Europe is shown in Appendix A. 

### 3.9. Asia and Africa

Fifteen seroprevalence studies were from five countries in Asia, namely Turkey (8), Mongolia (3), South Korea (2), Japan (1), and Malaysia (1), and two studies were from Africa (one each for Djibouti and Togo) (Appendix A). Aside from the study in Hokkaido, Japan, all of the studies in general and high-risk populations from Asia and Africa were conducted in non-endemic TBE areas, with anti-TBEV seroprevalence ranging between 0 and 20.2%. However, excluding data from studies in Mongolia (2) considered to be of “low” quality, the anti-TBEV seroprevalence in Asia and Africa ranged between 0 and 6%. The aggregated anti-TBEV seroprevalence across the region was 2.6% (95% CI: 0.8–5.1%; I^2^ = 97%) for general populations and 1.7% (95% CI:0–7.3%; I^2^ = 96%) for high-risk populations (Appendix A).

The eight studies in Turkey tested specimens from 2002 to 2016. The participants in six of the studies were in general population groups (anti-TBEV seroprevalence min–max: 0–4.6%) and two were in high-risk population groups (anti-TBEV seroprevalence min–max 1.4–3%). The two studies in South Korea (2015–2018) measured 1.9% anti-TBEV seroprevalence in a group of farmers on Jeju Island between 2015 and 2018 and 0.3% in both farmers and foresters across the country between 2017 and 2018 [54,55]. The latter study identified autochthonous TBEV infections for the first time in South Korea. Finally, samples from military members in Japan (n = 291) were tested for TBEV antibodies and identified apparent autochthonous TBEV infections in Hokkaido prefecture in two (0.7%) anti-TBEV seropositive individuals in 2017 [56]. The mean seroprevalence of TBEV antibodies for each population group in Asia and Africa is summarized via a forest plot in Appendix A. 

## 4. Discussion

### 4.1. Summary of Results

This is the first comprehensive summary of anti-TBEV seroprevalence studies conducted globally from the 1950s to date. Our analysis found wide variations in anti-TBEV seroprevalence across regions, over time, and among study populations. As expected, the prevalence of anti-TBEV antibodies was higher among individuals with occupational exposure to ticks and in persons who reported frequent exposure to ticks than in the general population, and in studies from Northwestern Eurasia (Finland, Lithuania, and Russia) compared to other geographic regions.

Both methodological and epidemiological factors may explain the differences in TBEV seroprevalence among studies. Importantly, given the sensitivity of TBEV serological tests, studies that do not use serological methods capable of confirming anti-TBEV antibody specificity are likely to overestimate anti-TBEV seroprevalence. This may have contributed to the higher seroprevalence observed in studies in Northwestern Eurasia, which was the region that had the lowest proportion of studies that used a confirmatory anti-TBEV antibody diagnostic method and where other orthoflaviviruses like WNV are endemic, further increasing the measured anti-TBEV seroprevalence if non-specific assays are used [57]. We note, however, that the reported human incidence of TBE in Lithuania is the highest across all of Europe; thus, the reported seroprevalence rates may be accurate [3].

Seroprevalence studies provide an estimate of the prevalence of past and current TBEV infections and can be used to estimate the incidence of TBEV infections based on the mean duration of antibody persistence. Seroprevalence studies conducted in countries with TBE-endemic areas can provide data to support public health interventions, including educating the public on tick prevention and TBE vaccination. 

The seroprevalence study participants in the high-risk population group were especially diverse. Most were foresters and agricultural workers, but others included military personnel (Lithuania, Japan, and Bosnia and Herzegovina), shepherds (Turkey), orienteers (Sweden), and muskrat catchers (the Netherlands). Since these outdoor occupations likely result in frequent exposure to tick populations, it is unsurprising that the prevalence of TBEV antibodies in high-risk cohorts was higher than that in general populations. Interestingly, we observed significant decreases in the prevalence of TBEV antibodies in both population groups (3.9-fold in high-risk populations) from before to after 1992, which may be related to the change in serological methods over time across the studies. Studies prior to 1992 more often used serological methods with less specificity to detect TBEV antibodies, which could have resulted in the overestimation of the anti-TBEV seroprevalence due to cross-reactivity with antibodies from other orthoflaviviruses. Further, most (83%) of the participants in our analyses were considered “unvaccinated” against TBE, and “vaccinated” participants were excluded entirely unless NS1-ELISA serology was utilized, which may have reduced the representativeness of the study populations. We observed a similar decrease (2.3-fold) among high-risk population groups (but not general populations) from TBE-endemic countries only. The reasons for the decline in TBEV antibody prevalence in high-risk populations are not entirely known but may be related to surveillance or sample bias in seroprevalence studies conducted before 1992 that targeted high-risk population study participants from high-incidence TBE or “high-risk” TBE areas within TBE-endemic countries. Overall, however, we observed no difference in anti-TBEV seroprevalence among study participants across both population groups within “high-risk” TBE areas from TBE-endemic countries between the two time periods, which is consistent with the increasing TBEV exposure and incidence of TBE due to natural infections [58]. 

Global climate change and socioeconomics/politics are other factors that may partially explain the observed divergent effect between the increased number of reported TBE cases and decreased anti-TBEV seropositivity in some population groups in this study. Climate change is frequently associated (both directly and indirectly) with increased TBE incidence due to its multifactorial impact on ecological, host, tick, and human factors involved in the natural cycle of TBE [59]. Increasing temperatures over the past three decades in Europe have broadened the suitable geographic and altitudinal habitats for *Ixodes* spp., and the milder winters in recent years have generally resulted in greater abundances of *I. ricinus* nymphal populations, as documented by long-term tick surveillance [60,61]. Mathematical and mechanistic models largely agree on the climate’s catalytic impact on tick phenology across life cycles and generations but are mixed regarding the downstream outcomes for enzootic cycling of the virus, with some predicting greater tick abundances and increased TBEV transmission within current endemic regions such as Hungary [62] and others suggesting less efficient and asynchronous co-feeding activities between tick larvae and nymphs on competent reservoir hosts [63,64]. Due to the complex interplay with tick/host populations, animal virus reservoirs, and seasonal activities, climate change may indirectly (but positively) influence the interannual cyclical oscillations of TBEV exposure periods, lengthening the periods for TBEV transmission [65,66]. However, although climate change is commonly implicated with increases in both the abundances of new TBEV foci and reported TBE cases in Europe, other factors, such as changing socioeconomic and political policies by national governments, especially in Northwestern Eurasia, can strongly impact potential TBEV exposure in humans (and subsequent anti-TBEV seropositivity and TBE incidence) by altering human behaviors and outdoor activities [67,68]. These policies, over the past two decades and throughout history (e.g., Cold War), have likely greatly influenced human surveillance activities and reports and further explain the inverse results that we observed in this review between the prevalence of TBEV antibodies across some population groups and the number of reported TBE cases over time in Europe. 

TBEV infection can lead to hospitalization and serious health outcomes, including long-term sequelae and death. According to the World Health Organization and ECDC, vaccination is the most effective way to prevent TBEV infection [58,69]. Public availability of and access to TBE vaccines has increased broadly across many European countries following the introduction of FSME-Immun^®^ to the Austrian public in 1981 and the launch of Encepur^®^ in Germany in 1991 [2,70], with notable success in reducing the number of TBE cases below one case per 100,000 population/year over the last ten years in Hungary [71] and a documented 99% field effectiveness against hospitalized cases in Austria, although the reported TBE cases have increased in the last decade, even though vaccine uptake is >80% based on a survey study [72,73,74]. Many TBE-endemic countries recommend, and some countries reimburse, TBE vaccination for persons in high-risk population groups. In Switzerland and Austria, for example, TBE vaccination is recommended nationally for everyone and provided free of charge to individuals with occupational risks or who engage in high-risk behaviors that result in tick exposure [75,76]. In Estonia, children as young as one year of age can receive the TBE vaccine, and Slovenia partially funds TBE vaccine doses for children born in 2016 or after and adults born in 1970 or before [77,78]. Although numerous studies have demonstrated the high effectiveness of European TBE vaccines, including FSME-Immun^®^ and Encepur^®^, vaccine coverage is often low in endemic countries, even in those like Germany, where the vaccine is widely available and provided free of charge [70,79]. A recent seroprevalence study in Baden-Württemberg, Germany identified an anti-TBEV antibody prevalence of 5.7% using the NS1-ELISA and concluded that the incidence of TBEV infections in the general population has increased over the past five decades, highlighting the need for increased vaccine uptake in the country [21,39]. 

Several serological test methods for the measurement of anti-TBEV antibodies, with varying sensitivity and specificity, are available for use in TBE seroprevalence studies. The most frequently utilized test method in the studies in this review was the conventional ELISA test; however, this test did not become the predominant method until after 1992. Before 1992, HI assays were most widely used. Although a conventional IgG ELISA or HI test is useful for the initial screening of specimens for the presence TBEV antibodies, a second confirmatory test, such as an NT or NS1-ELISA, should be used to increase the specificity of anti-TBEV antibody detection, since conventional IgG TBEV ELISAs possess highly variable anti-TBEV antibody specificity [80]. Importantly, failing to use a confirmatory serological test may result in false positives, leading to an overestimated seroprevalence. This is particularly true in non-endemic TBE areas with lower TBE incidence or in areas with co-circulation of multiple orthoflaviviruses, such as WNV, Usutu Virus (USUV), DENV, or JEV, and in countries that have widespread DENV and JEV vaccination programs [81,82,83]. For example, a seroprevalence study of human IgG antibodies against TBEV in Vietnam, where JEV and DENV are both highly endemic and JEV vaccination is widely administered throughout the country, reported that nearly half (47.4%) of the study participants were seropositive for TBEV antibodies, as measured by IIF testing [82]. NS1-ELISA testing is an important new tool as it can confirm the presence of TBEV (like an NT) and be conducted under biological safety level (BSL)-2 laboratory conditions, while NT testing requires BSL-3 containment facilities. Importantly, the NS1-ELISA test can distinguish between anti-TBEV antibodies produced by natural TBEV infection and those produced by TBE vaccination and therefore facilitate the interpretation of anti-TBEV seroprevalence results in populations with substantial vaccine uptake [14,19,21,84]. As evidenced in our synthesis, we excluded a substantial subset of extracted anti-TBEV seroprevalence data from pooled meta-analyses in instances where we could not infer whether the antibodies were specific to TBEV antigens (categorized as “low”-quality study data) to reduce data uncertainty. 

### 4.2. Utility and Considerations of Seroepidemiological Cohort Studies for TBE

The utility of a TBE seroprevalence study to provide useful metrics or indicators that best reflect the TBE risk and/or burden of disease is dependent on the study’s ability to collect reliable seroprevalence data relative to the local epidemiological context for TBE. Additionally, the serological method(s) used in the study should be selected to provide the most accurate data. The potential pitfalls and implications of some epidemiological variables for a TBE seroprevalence study in a population group are described (Table 4). The epidemiological variables that we considered for the data syntheses in this review were based on (i) the TBE vaccination status and medical/travel history of the study participants; (ii) the potential for local co-circulation of TBEV with other members of *Orthoflavivirus*; and (iii) the capacity of the serological method(s) used in the study to reliably detect anti-TBEV antibodies and accurately measure seroprevalence. As an example, an NT could be sufficient for a TBE seroprevalence study in Belgium for the purposes of determining the potential geographic expansion of TBE-endemic areas, since vaccine coverage rates are low and the risk of detecting anti-TBEV seropositive subjects from TBE vaccination is minimal. If this hypothetical study were unable to confirm the medical history (e.g., travel and vaccination status) in seropositive samples or individuals, an NS1-ELISA could verify a positive detection of TBEV antibodies caused by local autochthonous transmission and indicate the potential expansion of TBE. 

Lastly, additional epidemiological information at the local level could also be considered before conducting a TBE seroprevalence study, including (i) an assessment of human surveillance networks to identify potential surveillance gaps, biases, and underreporting of TBE cases; (ii) potential differential vaccine uptake across demographic groups; and (iii) an understanding of the local TBE ecology that influences the environmental tick hazards for local populations and/or the presence of TBEV foci (e.g., anti-TBEV seropositive mammalian hosts). The appropriate serological method depends on each of these factors. Additionally, the serological method(s) used in the study should also consider the study scope and study design to collect the most reliable seroprevalence data that best informs TBE epidemiology, including the following:Estimate travelers’ risk of exposure to TBEV through pre- vs. post-travel anti-TBEV seroconversion;Describe temporal trends in TBEV infections and compare clinical outcomes across cohorts using prospective cohort or longitudinal studies;Identify risk factors associated with TBEV infections through a case–control approach (e.g., high vs. low-risk, endemic vs. non-endemic, urban vs. rural, children vs. adults);Assess neutralizing antibody protection or potential vaccine coverage in a population or geographic area;Understand the baseline exposure rates to TBE in non-endemic populations to assess if TBE is an emerging or growing public health problem;Estimate the disease burden of TBE by monitoring the prevalence of TBEV antibodies relative to the number of reported clinical cases.

### 4.3. Limitations

The heterogeneity of seroprevalence studies contributed to several limitations in this review, particularly the differences in the serological methods used to identify anti-TBEV antibodies. We observed high levels of data heterogeneity in the meta-analyses (I^2^ > 90%), which is typical when aggregating data from multiple studies over long time periods [85]. Differences in the test sensitivity and specificity (and manufacturers if applicable) of the seven serological methods used in the studies may explain some of the observed differences in seroprevalence. Notably, studies that did not control for co-circulation or previous exposure to multiple orthoflaviviruses (WNV, USUV, DENV, JEV, etc.) may have overestimated anti-TBEV seroprevalence due to serological cross-reactivity, which would bias our meta-analysis. Many studies did not report the TBE vaccination histories of the study participants, and, for those that did, there were marked differences in the approaches to ascertaining vaccine histories. Inferring the vaccination status of the study cohort was often challenging if the studies did not confirm via serology or questionnaires relative to when (vaccine introduction; endemicity of other flaviviruses) and where (TBE endemicity) the study occurred. Vaccine coverage data at the country level or by age cohort, risk group, and year are rarely available. Further, some study cohorts or study subjects were poorly defined. In these instances, the subjects were labeled as general populations since cohorts like “healthy donors” and “blood bank sera” presumably included high-risk-group subjects as part of a broad representative population. Given the uncertainty of TBE vaccine histories and cohort populations with differential risks for TBE, the quality of anti-TBEV seroprevalence studies will improve with the wider availability of the NS1-ELISA testing method. Additionally, the completeness of the description of the seroprevalence study design and laboratory methods varied between studies, with older studies commonly providing the least information. We did not perform an in-depth systematic qualitative assessment of the included studies in this review and a data-quality-based threshold for inclusion was not used. For these reasons, the seroprevalence estimates from the most recent seroprevalence studies, particularly those that used the NS1-ELISA test method, likely more accurately reflected the true TBEV seroprevalence. If we were unsure whether the seroprevalence data were specific to TBEV, we tried to objectively categorize the data for inclusion or exclusion appropriately.

## 5. Conclusions

The overall seroprevalence of antibodies against TBEV due to natural infection in the participants of seroprevalence studies conducted in 33—mostly TBE-endemic—countries between 1956 and 2022 was 2.9–4.6%. There was an observed decline in anti-TBEV seroprevalence in participants in high-risk population groups, possibly associated with the utilization of more specific serological testing methods to detect anti-TBEV antibodies, but no difference in anti-TBEV seroprevalence among study participants from “high-risk” TBE-endemic areas overall. TBEV seroprevalence studies have been conducted with different serological testing methods that have varying sensitivity and specificity to reliably detect and measure the prevalence of TBEV antibodies in humans. The recent availability of the NS1-ELISA test method will improve the interpretability of seroprevalence results by distinguishing antibodies resulting from infection or vaccination. The harmonization in the study designs and test methods for TBEV seroprevalence studies would enhance cross-study comparisons and enable the improved targeting of TBE vaccination and prevention efforts. Collectively, this review emphasizes the urgent need for internationally accepted and standardized procedures when collecting seroepidemiological data for TBE that verify the serological methods with established diagnostic thresholds for the accurate detection of anti-TBEV antibodies, the medical histories of the study participants indicating prior vaccinations for TBE, and the potential exposure to or local presence of other orthoflaviviruses in the study region(s), in order to reliably assess and compare the potential risks of TBE or the burden of the disease over time and across countries.

## Figures and Tables

**Figure 1 vaccines-12-00854-f001:**
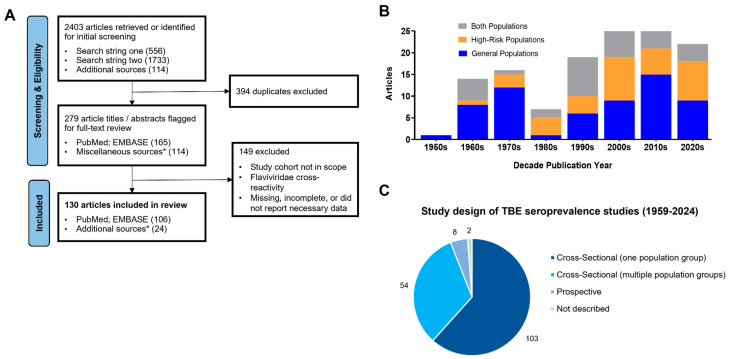
Characteristics of the 130 articles included for review on human seroprevalence for TBEV antibodies published between 1958 to 2024. The flow diagram for the retrieval, screening, and review of TBE seroprevalence studies in PubMed and Embase databases and additional sources is shown (**A**). The number of articles published over time was summed by decade (**B**). The proportions of TBE seroprevalence studies that were cross-sectional with one population group, cross-sectional with multiple population groups, prospective, or not described were calculated (**C**). * Additional and miscellaneous sources were defined as articles identified by grey literature or snowballing as described in the Methods.

**Figure 2 vaccines-12-00854-f002:**
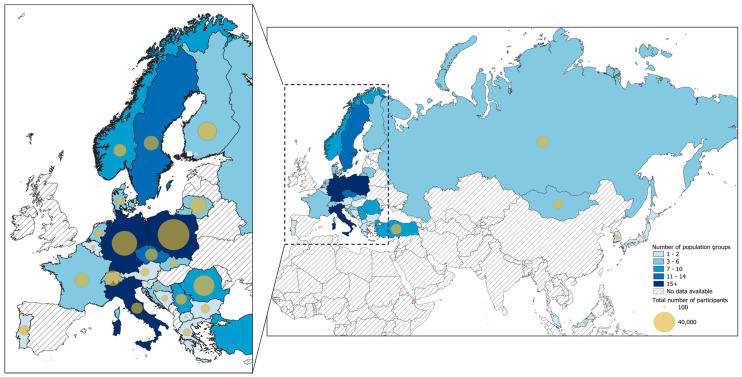
Map illustrating the total number of population group studies and study participants included for review by country, 1956-2022. Complete details are provided in Appendix A.

**Figure 3 vaccines-12-00854-f003:**
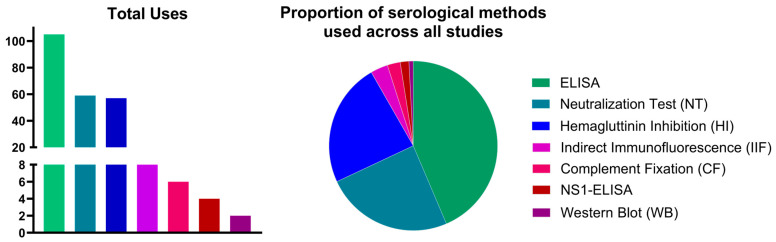
Summary of the serological methods used across all population group TBE seroprevalence studies. Each time a serological method was used to detect TBEV antibodies in humans and measure anti-TBEV seroprevalence in a study was recorded and calculated in aggregate (bar graph) and compared proportionally across all studies (pie chart). No data were excluded for serological method summaries.

**Figure 4 vaccines-12-00854-f004:**
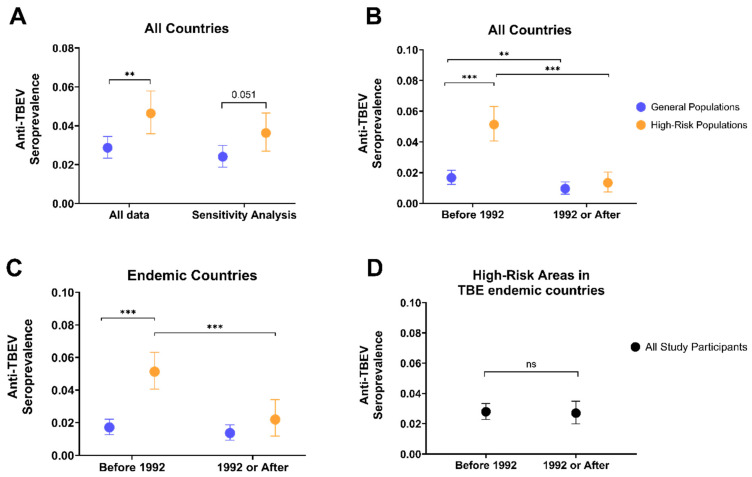
Overall anti-TBEV seroprevalence within high-risk populations is higher than general populations via meta-analysis. The prevalence of TBEV antibodies for general populations and high-risk populations across all countries (both endemic and non-endemic) and all studies (left column) or all countries and studies considered at least “moderate-quality” (right column) (**A**). The pooled seroprevalence before and after 1992 for each population group across all countries (**B**), TBE endemic countries (**C**), or in high-risk areas within TBE endemic countries across all study participants (**D**) was determined using “moderate” and “high” quality study data only. ** *p*-value =< 0.01; *** *p*-value =< 0.001. “ns” = not significant. Observations for (**A**) “Sensitivity Analysis” and (**B**–**D**) were determined using data from “moderate-quality” and “high-quality” seroprevalence studies only.

**Table 1 vaccines-12-00854-t001:** Categorization criteria for the qualitative assessment of the anti-TBEV seroprevalence data for TBE seroprevalence studies.

Study Quality	Minimum Study Criteria Thresholds ^1^	Included Analyses
High	NS1-ELISA Neutralization test; confirmation of medical and vaccination history (either reported in the study or inferred)	RE models Temporal analyses Summary statistics
Moderate	Neutralization test; unconfirmed medical or vaccination history Conventional ELISA; confirmation of medical and vaccination history; low likelihood for co-circulation of flaviviruses	RE models Temporal analyses Summary statistics
Low	Conventional ELISA; unconfirmed medical or vaccination status and/or higher likelihood for co-circulation of flaviviruses	RE models Descriptive analyses Summary statistics

^1^ Studies were categorized as high-quality, moderate-quality, or low-quality based on a qualitative assessment of the anti-TBEV antibody sensitivity and specificity of the serological methods used and the reliability and accuracy of the reported anti-TBEV seroprevalence among population groups. Criterion thresholds among population group studies considered which serological method(s) were used in the study relative to the known medical and TBE vaccination history of the study participants (known or unknown) and the potential circulation of multiple orthoflaviviruses near the study site of sample collection relative to the year(s) when the samples were collected.

**Table 2 vaccines-12-00854-t002:** Summary of serological methods used for TBE seroprevalence studies in general population and high-risk population groups. The total number of times that each serological method was used to measure TBEV antibody prevalence in a population group (1956–2022) before 1992, in 1992 or after, and across all studies was calculated. The proportion (%) for each serological method used in the studies is based on the total usage within each period (column).

Serological Method	before 1992 (%)	1992 or after (%)	Total (%)
ELISA	14 (16.7)	91 (58)	105 (43.6)
Neutralization Test (NT)	21 (25)	38 (24.2)	59 (24.5)
Hemagglutinin Inhibition (HI)	43 (51.2)	14 (8.9)	57 (23.6)
Indirect Immunofluorescence (IIF)	0 (0)	8 (5.1)	8 (3.3)
Complement Fixation (CF)	6 (7.1)	0 (0)	6 (2.5)
NS1-ELISA	0 (0)	4 (2.5)	4 (1.7)
Western Blot (WB)	0 (0)	2 (1.3)	2 (0.8)
Total	84	157	241

**Table 3 vaccines-12-00854-t003:** Overall summary of anti-TBEV seroprevalence across all studies and countries within general population and high-risk population groups (1956–2022).

Population Group (Participants)	Anti-TBEV Seroprevalence [95% CI]	I^2^	tau^2^	No. RE Model Observations	Data Range (Years)
All data; no exclusions					
General Population (114,973)	0.029 [0.023; 0.0355]	96.1%	0.019	354	1956–2022
High-Risk Groups (53,641)	0.046 [0.036; 0.058]	96.7%	0.020	148	1958–2022
Sensitivity analyses (high- and moderate-quality study data)					
General Population (89,599)	0.024 [0.019; 0.030]	95.6%	0.016	259	1956–2022
High-Risk Groups (44,422)	0.036 [0.027; 0.047]	96.0%	0.016	117	1958–2022

**Table 4 vaccines-12-00854-t004:** Potential sources of bias based on local epidemiological or study design factors for TBE seroprevalence studies. Potential sources of bias are related to the study area of interest and individuals or population group. Note: potential sources of bias may be applicable for several epidemiological factors.

Epidemiological or Study Design Variables	Potential Source(s) of Bias	Mitigations
TBE vaccination	Presence of TBEV antibodies from both TBE vaccination and natural infection (infected tick bite or consumption of contaminated dairy products)	Serological methods that distinguish between naturally acquired and vaccine-induced antibodies (NS1-ELISA) Accurate reporting of individual-level vaccination status or estimation of cohort-level vaccine uptake
Co-circulation of multiple orthoflaviviruses	Prior exposure (locally or travel-related) to other/multiple orthoflaviviruses	Serological methods that distinguish between TBEV and other *Orthoflavivirus* antibodies (NS1-ELISA; neutralization test)
Serology with poor sensitivity/specificity	Cross-reactivity with antibodies from non-TBEV orthoflaviviruses or TBE vaccination False negative/positive detection of anti-TBEV antibodies	Serological methods that distinguish between TBEV, other *Orthoflavivirus* antibodies, and TBE vaccination (NS1-ELISA; neutralization test)

## Data Availability

The original contributions presented in the study are included in the article/Appendix A; further inquiries can be directed to the corresponding author.

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
