# Peer review of "Global Seroprevalence of Tick-Borne Encephalitis Antibodies in Humans, 1956–2022: A Literature Review and Meta-Analysis"

_vaccines, 2024, doi:10.3390/vaccines12080854_

Round 1

Reviewer 1 Report

Comments and Suggestions for Authors

This is an interesting and important review specially for the human health services in countries (and regions) where Tick born enceplalites virus (TBEV)  are still prevalent, which includes most of Europe, Asia and Europe.  The figures are easy to understand.  

The only minor suggestion is to reduce the title of Table 1. Only the first sentence of the present figure would be enough. The  other part could be used as a footnote for the table. 

Author Response

Thank you for your suggestion. We have revised the title of Table 1 to include only the first sentence. 

Reviewer 2 Report

Comments and Suggestions for Authors

The authors present an important global meta-analysis concerning the seroprevalence of Tick-Borne Encephalitis Antibodies in Humans during the period 1956–2022. This work was well conducted, and the authors were careful with the methods to retrieve the information and with the data analysis. The global anti-TBEV seroprevalence during the studied period was 2.9-4.6%. I consider that this work will add new information in the area for the scientific community.

Author Response

Thank you very much for taking the time to review this manuscript. As no revisions/comments were requested from Reviewer 2, we left this page intentionally blank.

Reviewer 3 Report

Comments and Suggestions for Authors

Kelly, P.H. et al.: Global Seroprevalence of Tick-Borne Encephalitis Antibodies in Humans, 1956–2022: A Literature Review and Meta-Analysis

The authors have gathered and analysed an imposing volume of literature reporting on surveys of antibodies against TBEV conducted in Europe and beyond during the past 70 years. Literature sources inclusion was subject to availability in a digital form - which was, apparently, not the case in most of the Russian literature – whereby it may have happened that it is notably underrepresented (10 Russian references, contrast, eg., with 28 Turkish ones..!; Suppl.File 2).  Even so, to this referee’s knowledge, this is by far the most comprehensive review of TBE seroprevalence compiled to date, with the proviso that the epithet “global” in the title should be moderated. Congratulations are in order to the authors for their potentially highly impactful work. Prior to publication, however, the article should be amended as explained bellow:

 Of all issues identified in this manuscript, my greatest concern is about mechanical - I don’t hesitate to call it brainless – application of political regions [23] for geographical segmentation of the seroprevalence data (even if only for descriptive purposes). It seems that of all schemes at hand, the authors picked that least respecting epidemiological reality - a scheme, that, for example, puts into the same bag incompatible data that originated in Ixodes ricinus and I. persulcatus-endemic areas separated by thousands of kilometres (p.12, l.89-101), while, at the same time, sets apart countries the endemic areas of which form trans-boundary continua (e.g. Slovenia from Austria, Bavaria from the Czech Rep., ditto from Austria, etc.), where the sero-surveys in question were often conducted not further from each other[PZ1]  than a hundred, or two km. I don’t see any advantage of  such a regionalisation for this study (except for drastic simplification of the analysis) – I wish the authors have opted for another scheme, say the CIA World Factbook (https://commons.wikimedia.org/wiki/File:Regions_of_Europe_based_on_CIA_world_factbook.png), that is also a ‘political’ one yet acceptably matching TBE zonation. After all, the authors could design their own scheme compliant with the distribution of vectoring ticks and TBEV lineages; J.V.-Born.Dis.57:14??).

One puzzling (yet potentially important) outcome is that, for high-risk populations, the authors demonstrated decreasing trend in the prevalence of anti-TBEV antibodies contrasting with no change in the general population (p.10,l.1-19; Suppl.File 8), and hypothesise it is due to a qualitative shift in serological methods. Indeed, in all instances shown in Supplemental File 8, seroprevalence exhibits some, at least minimum decrease attributable to elimination of some false positives. However, it falls short of explaining the fact that the trends diverge significantly despite the two populations were surveyed using analogical methods (Suppl.File 7). Furthermore, because in TBE (well below endemic stability) seroprevalence and morbidity present communicating vessels, the observed decreasing trend is, in fact, paradoxical given the contemporary rise in case numbers. At least a brief discussion addressing the trends in TBE seroprevalence in the context of global changes of climate and the disease’s upturn is desired.

Some minor issues:

P.2, l.93:  regional summaries

P.4, 51-2: “…Data categorization of TBE “endemic” countries, “high-risk” areas, and “low-risk” areas is shown in Supplemental File 2…” – “endemic countries” or “endemic districts/provinces”, what is pertinent ?  Note, that categorization of some countries is equivocal – for example,  populations/rows belonging to Italy are once labelled “endemic”, once “non-endemic”…, check it, pls.

P.14, l.200: born in 1970 or before

Fig.3: Perfect the description of vertical axes, pls.

 [PZ1]

Author Response

Response to Reviewer 3 Comments

1. Summary

Point-by-point response to Comments and Suggestions for Authors

Comment 1: Of all issues identified in this manuscript, my greatest concern is about mechanical - I don’t hesitate to call it brainless – application of political regions [23] for geographical segmentation of the seroprevalence data (even if only for descriptive purposes). It seems that of all schemes at hand, the authors picked that least respecting epidemiological reality - a scheme, that, for example, puts into the same bag incompatible data that originated in Ixodes ricinus and I. persulcatus-endemic areas separated by thousands of kilometres (p.12, l.89-101), while, at the same time, sets apart countries the endemic areas of which form trans-boundary continua (e.g. Slovenia from Austria, Bavaria from the Czech Rep., ditto from Austria, etc.), where the sero-surveys in question were often conducted not further from each other[PZ1]  than a hundred, or two km. I don’t see any advantage of  such a regionalisation for this study (except for drastic simplification of the analysis) – I wish the authors have opted for another scheme, say the CIA World Factbook (https://commons.wikimedia.org/wiki/File:Regions_of_Europe_based_on_CIA_world_factbook.png), that is also a ‘political’ one yet acceptably matching TBE zonation. After all, the authors could design their own scheme compliant with the distribution of vectoring ticks and TBEV lineages; J.V.-Born.Dis.57:14??).

·        Response 1: Thank you for your candor in this comment and for proposing suitable alternatives for the regional groupings with respect to TBE. Lamentedly, in our attempt to provide sensible and more interpretable regional summaries, we failed to properly consider how the regional groupings via the United Nations Geoscheme lacked consistent epidemiological factors and ecological conditions for the enzootic nature of TBE. As such, we agree with your suggestion and have revised the regional groupings according to our own design scheme and loosely based on the suggested alternative of the CIA World Factbook. We explain our revised regional groups in the Methods (P4, lines 156-161), amended the Results Sections 3.5-3.9 accordingly based on redoing the meta-analyses under the new groupings, and updated Supplemental Files 2-5 with new labels and Supplemental Files 9-10 with new random effects model results and forest plots.  

Comment 2:  One puzzling (yet potentially important) outcome is that, for high-risk populations, the authors demonstrated decreasing trend in the prevalence of anti-TBEV antibodies contrasting with no change in the general population (p.10,l.1-19; Suppl.File 8), and hypothesise it is due to a qualitative shift in serological methods. Indeed, in all instances shown in Supplemental File 8, seroprevalence exhibits some, at least minimum decrease attributable to elimination of some false positives. However, it falls short of explaining the fact that the trends diverge significantly despite the two populations were surveyed using analogical methods (Suppl.File 7). Furthermore, because in TBE (well below endemic stability) seroprevalence and morbidity present communicating vessels, the observed decreasing trend is, in fact, paradoxical given the contemporary rise in case numbers. At least a brief discussion addressing the trends in TBE seroprevalence in the context of global changes of climate and the disease’s upturn is desired.

·        Response 2: Thank you for this comment as we also agree it is a perplexing situation and one that cannot be explained entirely by different serology methods over the past eight decades. Please find in P16-17, lines 243-271 substantial additions related to global climate change and socioeconomic/political policies that could further explain the divergent trends in reported TBE cases and anti-TBEV seroprevalence we observed, per your suggestion.

·        Response 2 (continued): Additionally, we also would like to bring your attention to Lines 230-242 in the Discussion where we comment on other factors in the original studies that may bias/impact the reported/observed anti-TBEV seroprevalence and results from analyses. We noted that in addition to the transition to using more sensitive and specific serological testing methods, the reported anti-TBEV seroprevalence may be also be artificially lower in more recently published studies (e.g., in 1992 or after as shown in Figure 4) compared to historical studies due to 1) the recruitment of more convenient versus random cohort subjects in the studies (Section 4.1; Lines 230-237); and 2) different study designs or study scopes that may have prioritized the selection of study regions considered “low risk” or “non-risk” TBE areas to assess the potential spatial spread of TBE into non-endemic regions (Section 4.1; Lines 238-242).  During the full-text reviews of included studies, it was evident that studies conducted prior to 1992 were more likely to be conducted in areas with high reported TBE incidences and/or include individuals who were at high-risk of exposure to TBEV-infected tick populations (e.g., occupational hazards). Presumably, it is likely that these historical studies targeted the recruitment of the most at-risk populations within TBE endemic regions to rapidly understand potential local risks of a life-threatening disease at a time when no public health interventions such as a vaccine were widely available to the public.

·        Response 2 (continued): Lastly, we would like to point out that we attempted to further explain the potential bias in the divergent trends in reported TBE cases and anti-TBEV seroprevalence by comparing the anti-TBEV seroprevalence among the subjects residing in TBE endemic areas in aggregate (regardless of population group) before and after 1992 and found no difference between the two periods (Figure 4D; Discussion; Lines 239-242). These results, controlling for TBE endemicity versus cohort status, may more accurately reflect the actual (and consistent) prevalence of anti-TBEV antibodies across human cohorts in endemic areas over time and partially explain the increase in TBE incidences over time. We also make note how our exclusion criteria that omitted data from subjects with unknown vaccination history (Discussion; Lines 230-232) or likely exposure to other orthoflaviviruses also contributed to data uncertainty (Discussion; Lines 320-323)

Comment 3: P.2, l.93:  regional summarie

·        Response 3: We have amended P.2, lines 92-93 to address your revision as shown below (Note: Deleted text is shown as strikethrough and text added to the manuscript is shown in bold and underlined in red):

“Accurate regional estimates of the TBEV seroprevalence can also be difficult in areas regions with even modest vaccination coverage…”

Comment 4: P.4, 51-2: “…Data categorization of TBE “endemic” countries, “high-risk” areas, and “low-risk” areas is shown in Supplemental File 2…” – “endemic countries” or “endemic districts/provinces”, what is pertinent ?  Note, that categorization of some countries is equivocal – for example,  populations/rows belonging to Italy are once labelled “endemic”, once “non-endemic”…, check it, pls.:

·        Response 4: Thank you for this question which is important to clarify for proper interpretation of the meta-analysis. We believe both categorization schema (country endemicity status and risk areas) are critical for data comparisons and analyses. Country endemicity for TBE was based on the countries’ local “recurrent annual autochthonous transmission of TBEV infections over multiple years at the time (year) of sample collection” (P4, lines 161-167). Therefore, countries such as Italy could be differentially categorized for TBE endemicity over time as the virus has established its geographic reach over time when the studies were conducted. Country endemicity status was an important variable to aid broader cross-country data comparisons with more similar study designs and sample cohorts. The additional spatial categorization data collected from subjects residing in high risk (endemic regions within TBE endemic countries) and low risk (non-endemic regions within TBE endemic countries) provides further stratification to compare data similar data across studies.  

Comment 5: P.14, l.200: born in 1970 or before

·        Response 5: Thank you for catching this error. We have made the necessary correction to state “before” instead of “after” in the sentence (Line 287).

Comment 6: Fig.3: Perfect the description of vertical axes, pls.

·        Response 6: Thank you for this comment as the figure legend was confusing. We have revised the Figure Legend as suggested to the following:

“Summary of the serological methods used across all population group TBE seroprevalence studies. Each time a serological method was used to detect TBEV antibodies in humans and measure anti-TBEV seroprevalence in a study was recorded and calculated in aggregate (bar graph) and compared proportionally across all studies (pie chart). No data were excluded for serological method summaries.”

5. Additional clarifications

No additional comments or clarification

Reviewer 4 Report

Comments and Suggestions for Authors

Kelly et al. gathered TBEV serological data available online of scientific papers from the whole world published over the last almost seven decades. Such studies are supportive in writing reviews and to have a broad view of the TBE situation worldwide. Although we should mention (and the authors should also mention it in their manuscript) that comparing results of serological data made by not unified, validated detection methods, assays, from countries thousands of km-s and decades from each other is not possible. Similarly any drawn conclusions from such a work could be (are) misleading. These serological values, numbers could mean only a hint about the real (continuosly changing) TBEV situation.

-        line 46. In Europe it is true. In the Far East 30-40% of clinically ill people die.
-        lines 189-207 – In Hungary vaccination (like in Austria) completely solved the TBE problem. Number of the annual diagnosed TBE human cases from 
250-350 (1980-96) went down to 10-15 in 20 years. Egyed et al., Zoon. Publ. Health 2023. 70, 81-92.

-        lines 208-215. How could we deduce any conclusions from such a wide variety of serological methods applied in various countries over decades?

-        4.3 Limitations line 280. Situation of related flavivirus infections, endemic areas (JEV, WNV, Usutu +50 others) also disturb the evaluation the results of TBEV serological tests.

-        The authors should underline somewhere (discussion, conclusions) that internationally accepted verification of diagnostic methods, data recording of participants, indicating vaccination background, data about the presence of other flaviviruses in the tested regions are inevitable for reliable analysing and comparing results of TBEV serological tests carried out in different countries.

Figure 4 indicates (at least for me) that seropositivity has been decreased in whole countries (endemic and not endemic regions) over the past decades. What could be the reason for that?

I suggest two maps, one for Europe and one for Asia (as TBE is problem of Eurasia) where numbers of studies and minimal-maximal detection serological values are indicated (in %) on the affected countries, where from at least 5 studies were available.

Comments on the Quality of English Language

 spelling:

-                      line 133 population(s) groups (repetiation in the next line)

- line 200 - 65]spaceAlthough        a space is missing

Author Response

Response to Reviewer 4 Comments

1. Summary

Point-by-point response to Comments and Suggestions for Authors

Comment 1: line 46. In Europe it is true. In the Far East 30-40% of clinically ill people die.

·        Response 1: Thank you for pointing this out. We added additional text in the Introduction on the three main TBEV subtypes (Eu, Sib, and FA) and specified the case fatality rates associated with each clarity on P2, lines 45-52.

Comment 2: lines 189-207 – In Hungary vaccination (like in Austria) completely solved the TBE problem. Number of the annual diagnosed TBE human cases from 250-350 (1980-96) went down to 10-15 in 20 years. Egyed et al., Zoon. Publ. Health 2023. 70, 81-92.

·        Response 2: Thank you for raising this important example and providing the citation. We added specific examples of the TBE vaccines’ success reducing TBE cases in both Hungary and Austria as requested in the Discussion, Section 4.1, Lines 277-280.

Comment 3: lines 208-215. How could we deduce any conclusions from such a wide variety of serological methods applied in various countries over decades?

·        Response 3: Thank you for this question as it is complex and requires nuance when answering in terms of TBE seroepidemiology. In short, it would not be appropriate to deduce broad conclusions into the spatiotemporal risks/burden of TBE based on the prevalence of TBEV antibodies alone due to myriad factors described at length in the manuscript most notably varying and unknown medical histories among study participants, serological cross-reactivity with other antibodies from other orthoflaviviruses, and data biases due to variable surveillance, study heterogeneity, and reporting. For these reasons and in order to infer potential conclusions related to anti-TBEV seropositivity, we created a qualitative assessment and data categorization criteria (described in Table 1) to more reliably compare seroepidemiological data across countries and over time that employed similar methods, designs, and provided more complete information related to the study (e.g., participant medical histories).

Comment 4: 4.3 Limitations line 280. Situation of related flavivirus infections, endemic areas (JEV, WNV, Usutu +50 others) also disturb the evaluation the results of TBEV serological tests

·        Response 4: Thank you for emphasizing this important point. Although this is highlighted numerous times throughout the manuscript, we have also added this aspect to Section 4.3 Limitations Lines 376-379 as suggested.

Comment 5: The authors should underline somewhere (discussion, conclusions) that internationally accepted verification of diagnostic methods, data recording of participants, indicating vaccination background, data about the presence of other flaviviruses in the tested regions are inevitable for reliable analysing and comparing results of TBEV serological tests carried out in different countries.

·        Response 5: This is a great idea. We have added a sentence emphasizing this key point at the end of the Conclusions, Lines 414-420.

Comment 6: Figure 4 indicates (at least for me) that seropositivity has been decreased in whole countries (endemic and not endemic regions) over the past decades. What could be the reason for that?

·        Response 6: Thank you for this question as it is an important topic in terms of TBE epidemiology and data interpretation. To clarify, we defined a country as a TBE “endemic” country according to the World Health Organization which states a “endemicity is defined as having at least one TBE endemic region within the country which requires local circulation and human transmission of the TBEV and a reported TBE case.” Therefore, the decreased anti-TBEV seropositivity over time in TBE endemic regions may not be occurring across whole countries but rather specific regions historically considered to be endemic for TBE.

·        Response 6 (continued): We believe we have addressed several reasons for the observed decrease in anti-TBEV seropositivity from our meta-analysis which are detailed extensively in the Discussion, Section 4.1, Lines 203-270). Briefly, these additional factors that may also partly explain the observed anti-TBEV seropositivity decrease include public health interventions introduced in the public such as vaccination, data bias due to varying study design/scope and differential cohort recruitment strategies (e.g., prioritization of study regions in non-endemic TBE areas to assess spatial spread of TBE which would have reduced anti-TBEV seropositivity among local cohorts), and increased sensitivity and specificity of newer serological methods which would reduce false positive samples compared to older studies which were more likely to use serology that were more likely to cross-react with antibodies against other orthoflaviviruses or from prior vaccination against TBEV. Importantly, we compared anti-TBEV seroprevalence among study participants across both population groups within “high-risk” TBE areas from TBE endemic countries between the two time periods and found no difference which is consistent with increasing TBEV exposure and TBE incidence due to natural infections and may be better explain the reported increase in TBE incidences over time (Section 4.1; Lines 239-242). We also added context related to climate change and socioeconomic policies (Section 4.1; lines 243-271) that may have impacted anti-TBEV seroprevalence and TBE incidence.

Comment 7: I suggest two maps, one for Europe and one for Asia (as TBE is problem of Eurasia) where numbers of studies and minimal-maximal detection serological values are indicated (in %) on the affected countries, where from at least 5 studies were available.

·        Response 7: Thank you for this suggestion. We have revised the map illustration in Figure 2 to include a global map that includes the population groups studies/participants in Asia and Africa countries included in the review with an inset map of the total number of population groups studies/participants in European countries included in the review. We attempted to include the minimal-maximal detection of serological values in the map but it was ultimately decided that the values were not legible and difficult to incorporate without confusing the reader as some countries required separate minimal-maximum values for general populations and high-risk population group studies. Therefore, we expanded on Supplemental File 5 and added additional columns that specify the minimal-maximal serological values within each population group and their respective study periods (range years).

4. Response to Comments on the Quality of English Language

Point 1 (spelling): Line 133 population(s) groups (repetiation in the next line)

·        Response 1: Thank you for catching this error. We have corrected the spelling to “population”

Point 2: line 200 - 65]spaceAlthough       a space is missing

·        Response 7: Thank you for catching this error. We have corrected the grammar.

5. Additional clarifications

No additional comments or clarification

Round 2

Reviewer 3 Report

Comments and Suggestions for Authors

Kelly, P.H. et al.: Global Seroprevalence of Tick-Borne Encephalitis Antibodies in Humans, 1956–2022: A Literature Review and Meta-Analysis, v.2

The geographical TBE’s zonation, that the authors propose, is fine and makes sense - I suggest only a minor cosmetic correction for the authors’ consideration: they staked out an (“I. persulcatus“) zone ranging from the Baltics and Finland to the west to Transuralic Russia/Siberia to the east that they term “Eastern Europe” (p.13, l.104). Perhaps, it could be renamed to “Northeastern Europe” (or, better, to “Northwestern Eurasia”) to better pair with “Southeastern Europe” (p14, l.120).

Otherwise, I acknowledge that all other issues have been amended to my satisfaction, and – being convinced of the study’s outstanding value - I gladly recommend it for publication.

Author Response

Thank you for this good idea to rename the regional groupings accordingly. We have changed all of the "Eastern Europe" title groupings to "Northwestern Eurasia" as requested in the following sections: P13, L104-105 and L109; P14, L118; P15, L169 and L175; P16, L232; P20, L421.